# A Novel Language Paradigm for Intraoperative Language Mapping: Feasibility and Evaluation

**DOI:** 10.3390/jcm10040655

**Published:** 2021-02-08

**Authors:** Katharina Rosengarth, Delin Pai, Frank Dodoo-Schittko, Katharina Hense, Teele Tamm, Christian Ott, Ralf Lürding, Elisabeth Bumes, Mark W Greenlee, Karl Michael Schebesch, Nils Ole Schmidt, Christian Doenitz

**Affiliations:** 1Department of Neurosurgery, Regensburg University Hospital, 93053 Regensburg, Germany; delin-wolfgang.pai@ukr.de (D.P.); katharina.hense@ukr.de (K.H.); christian.ott@ukr.de (C.O.); karl-michael.schebesch@ukr.de (K.M.S.); nils-ole.schmidt@ukr.de (N.O.S.); christian.doenitz@ukr.de (C.D.); 2Institute of Social Medicine and Health Systems Research, Otto von Guericke University Magdeburg, 39106 Magdeburg, Germany; frank.dodoo-schittko@ukr.de; 3Institute for Experimental Psychology, University of Regensburg, 93053 Regensburg, Germany; teele.tamm@ymail.com (T.T.); mark.greenlee@ur.de (M.W.G.); 4Department of Neurology and Wilhelm Sander-NeuroOncology Unit, Regensburg University Hospital, 93053 Regensburg, Germany; ralf.luerding@ukr.de (R.L.); elisabeth.bumes@ukr.de (E.B.)

**Keywords:** intraoperative language mapping, direct cortical stimulation, awake surgery, neuropsychological outcome

## Abstract

(1) Background—Mapping language using direct cortical stimulation (DCS) during an awake craniotomy is difficult without using more than one language paradigm that particularly follows the demand of DCS by not exceeding the assessment time of 4 s to prevent intraoperative complications. We designed an intraoperative language paradigm by combining classical picture naming and verb generation, which safely engaged highly relevant language functions. (2) Methods—An evaluation study investigated whether a single trial of the language task could be performed in less than 4 s in 30 healthy subjects and whether the suggested language paradigm sufficiently pictured the cortical language network using functional magnetic resonance imaging (fMRI) in 12 healthy subjects. In a feasibility study, 24 brain tumor patients conducted the language task during an awake craniotomy. The patients’ neuropsychological outcomes were monitored before and after surgery. (3) Results—The fMRI results in healthy subjects showed activations in a language-associated network around the (left) sylvian fissure. Single language trials could be performed within 4 s. Intraoperatively, all tumor patients showed DCS-induced language errors while conducting the novel language task. Postoperatively, mild neuropsychological impairments appeared compared to the presurgical assessment. (4) Conclusions—These data support the use of a novel language paradigm that safely monitors highly relevant language functions intraoperatively, which can consequently minimize negative postoperative neuropsychological outcomes.

## 1. Introduction

An awake craniotomy with language mapping in patients with brain tumors or metastasis in language-critical brain areas represents the gold standard for maximizing the resection of brain tumors and metastases in language-critical brain areas while minimizing the risk for postoperative functional deficits [1,2,3]. Both aspects improve postsurgical treatment responses, overall survival, and increase patients’ health-related quality of life [4,5].

Mapping the complete language capacity with its subfunctions during language testing with direct cortical stimulation (DCS) in the setting of an awake craniotomy can be challenging because language with its subfunctions, such as phonology, morphology, semantics, syntax, and grammar, is organized as a complex network of cortical areas and fiber bundles around the sylvian fissure in the language-dominant hemisphere. There is a huge diversity of intraoperative language paradigms, such as object naming [6,7,8], sentence completion [9], action naming [10,11,12], or verb generation [7,13,14,15,16]. Unfortunately, these tasks only cover single aspects of language processing and might therefore insufficiently reflect patients’ linguistic capacity and skills. Rofes and Miceli reported huge differences between different intraoperative language tasks [17]. The most popular tasks, namely, intraoperatively applied picture-naming tasks, are extremely sensitive to phonological retrieval but they demand minimal linguistical processing compared to other tasks, such as verb generation or sentence completion, and therefore might fail to address more complex grammatical, syntactic, lexico-semantic, or morphophonological language processes [17]. This is in line with presurgical functional magnetic resonance imaging results that show that picture-naming tasks are incapable of reliably identifying Broca’s area in healthy subjects [18]. Especially for patients with a frontal tumor, it might therefore be helpful to increase the sensitivity of the object-naming task by adding a grammatical component to it. This may increase the likelihood of detecting eloquent language regions that may currently be overlooked using lexico-semantic tasks. The lesion location might also influence the choice of intraoperative language paradigms; however, linguistic subprocesses (such as the production and perception of phonology, morphology, semantics, and syntax) are normally not bound to a single brain area but are rather organized as networks [19]. Therefore, some others suggest an individual language test battery with different language tasks [19,20,21], but this leads to increasing the surgery time and possibly to patients’ being overtaxed and exhausted. This might also bear the risk of false-positive language disturbances because of patients’ impaired attention [22].

Besides the validity of an intraoperative language task, the task must be designed to meet the technical demands of DCS by not exceeding the assessment time of 4 s in a single language trial to minimize the risk for DCS-induced seizures [10,12,23,24]. This also impacts the design of intraoperative language tasks because complex tasks are highly likely to need more than 4 s to be performed, even if those tasks might be able to cover more or even all linguistic subfunctions that are necessary for the generation of a correct language utterance compared to easy language paradigms, such as object naming or verb processing alone.

In this study, we aimed (1) to design and evaluate a novel single-language paradigm consisting of a combination of picture naming and sentences generation tasks that better reflect the highly relevant language functions and (2) to simultaneously restrain the assessment duration of single-task trials to up to 4 s to minimize intraoperative complications, such as seizures. Therefore, we investigated whether this language paradigm could be performed within 4 s in a healthy subject group to meet the criteria of DCS and whether the language task could stimulate the complete language system surrounding the sylvian fissure during a functional magnetic resonance imaging (fMRI) experiment. In the last step, (3) we investigated the novel language task intraoperatively in a group of brain tumor patients undergoing an awake craniotomy to test its feasibility and how it affected patients’ linguistical and cognitive outcome.

## 2. Materials and Methods

### 2.1. Language Paradigm

The language paradigm was designed as a single-stimulus trial comprising a combination of the classical picture-naming task using the DO (Test de dénomination orale d’images) 80 figures [25] and a semantically associated verb in the infinitive form. Both stimuli were presented visually in black and white by being randomly placed one below the other. We designed 80 different pairs of stimuli made up of a picture and a written verb. Healthy controls and patients were asked to generate grammatically and semantically correct sentences according to the given language stimuli (Figure 1). Linguistically, in the German language, this task comprised the correct choice of determiner genus, the processing of the correct word order, and a correct subject–verb congruence structure to build a sentence that follows the subject–predicate formula. The performance of this task thereby required the processing of phonemic/phonological, semantic, morphological, and syntactic, perceptive, and productive aspects of language per se, as well as the transfer of this information into a surface structure (speech).

### 2.2. Patient and Subject Sample

#### 2.2.1. Healthy Subjects—Evaluation Study

The first aim of the pilot study was to test the intraoperative feasibility of the language paradigm. This intended to evaluate whether the performance speed of the rather complex language task did not exceed the assessment time of 4 s in a single language trial to minimize the risk for intraoperative DCS-induced seizures. Therefore, 30 healthy native German speakers were included (16 female: mean age 26.0 years (*SD* = 8.49)).

The second aim of the pilot study was to investigate whether the suggested language paradigm could sufficiently picture the cortical language network, including the inferior frontal gyrus, the ventral premotor area, the angular and supramarginal gyri, and the posterior superior and middle temporal gyri in the language-dominant hemisphere. Therefore, 12 additional healthy right-handed native German speakers were included to participate in an fMRI experiment (6 female: mean age 27.7 (*SD* = 1.69)). These subjects did not take part in the former behavioral experiment (as described above) to avoid familiarity effects.

#### 2.2.2. Patient Group—Feasibility Study

The third aim of the study was the successful intraoperative application of the novel language task. Twenty-three right-handed patients and one left-handed patient without or with only mild presurgical language deficits with left lateral brain tumors in language-associated brain areas were included in the study between the years 2015 and 2019. The patients’ characteristics are shown in Table 1. An independent medical indication for awake surgery was given in all patients. The presurgical fMRI data gathered using a battery of verb and syntax generation tasks done by these patients showed a close spatial proximity of language critical areas and brain tumors. Left-hemispheric language laterality was computed in all patients. All patients underwent neuropsychological testing, including language tests, before and after surgery.

### 2.3. Procedures and Data Analysis

#### 2.3.1. Evaluation Study

To evaluate whether the performance speed of the novel language task exceeded the assessment time of 4 s in a single language trial, 30 healthy subjects were instructed to generate grammatically and semantically correct sentences according to the language stimulus in a time interval not exceeding 4 s. Stimulus presentation was done using the “Presentation” software [26]. The 80 language stimuli were subsequently presented and randomized for each subject. The mean subjects’ performance speed during each trial was measured by a supervisor who pressed a button immediately after each subject stopped speaking. Data analysis was done using IBM SPSS Version 25 (IBM Deutschland GmbH, Ehningen, Germany) by calculating the mean performance speed.

To investigate whether the suggested language paradigm could sufficiently picture most parts of the cortical language network, 12 healthy right-handed native German speakers (6 female; mean age 27.7 (*SD* = 1.69)) were instructed to perform the novel language paradigm while stimulus-dependent changes in their blood oxygen level-dependent (BOLD) responses were recorded. The language task was analogous to the behavioral paradigm design with respect to the stimulus presentation. Using a block design, five stimulus trials, each lasting 4 s, were presented sequentially and were followed by a resting period of 20 s. The data were collected using a 3T MRI scanner (Magneton Allegra, Siemens, Erlangen, Germany). The fMRI data parameter for T2*-images were 34 slices, repitition time (TR) = 2000 ms, echo time (TE) = 30 ms, voxel size = 3 × 3 × 3 mm^3^, flip angle = 90°, and field of view = 192 × 192 mm^2^. To visualize the results three-dimensionally, an additional T1 magnetization prepared rapid gradient echo (MPRAGE) image was acquired (TR = 2300 ms, TE = 2910 ms, flip angle = 9°, and slice thickness = 1 mm). Data analysis was done using the Statistical Parametric Mapping 8 (SPM8) software running (https://www.fil.ion.ucl.ac.uk/spm/ accessed on 7 February 2021) under Matlab 7.1 (Mathworks, Sherborn, MA, USA). Functional images were realigned, coregistered to the structural image, normalized to the Montreal Neurological Institute (MNI) space, and spatially smoothed using a full-width at half-maximum Gaussian kernel of 8 mm. Afterward, based on the general linear model, we defined a single regressor according to the language task timing profile, which was then convoluted using a boxcar model function with the canonical hemodynamic response function. The resting period was not modeled explicitly and served as an implicit baseline. A post hoc *t*-test was calculated for every voxel to compare the fMRI signals and beta-weights according to the stimulus regressor. The resulting individual data were the basis of a second-level random effects analysis. Voxels were defined to be significant if they did not fall below a *t*-value of *T* = 7 at the voxel level and simultaneously did not exceed a *p*-value of *p* < 0.001 (uncorrected, *T* = 4.025) at the cluster level. Only clusters with more than 50 voxels were reported.

#### 2.3.2. Feasibility Study

All 24 patients underwent an awake craniotomy without sedation following the wake–awake–awake technique with therapeutic communication that had been established by our group [27]. Due to internal administrative affairs, the pre- and postoperative neuropsychological data of only 13 patients were analyzed in this study. Neuropsychological assessment included the following cognitive domains: verbal and working memory (“number repeating” forward and backward spans) [28], non-verbal working memory (“Corsi block span”) [29], lexical and semantic verbal fluency [30], verbal [28] and visuo-constructive capacity, visual long-term memory (“Rey Visual Design Learning Test”) [31], and executive functions [32]. Long-term memory functions and visuo-constructive capacity were only assessed prior to surgery to avoid potential training effects. Patients were tested one day before surgery and 2–5 days after surgery. Performance differences between the pre- and postsurgical neuropsychological outcomes were computed by comparing the median percentile ranks of the single test scores with the nonparametric Wilcoxon–Mann–Whitney Test because percentile ranks probably do not fulfill the criteria of a normal distribution. The analysis was performed using IBM SPSS Version 25. The significance level was set to *p* < 0.05. Additionally, effect sizes were computed. To guarantee that patients could conduct a trial of the intraoperative language task within 4 s, all patients were intensively trained in advance to surgery. Accordingly, every single stimulus was first practiced by the patients. If patients were not familiar with some pictures shown during the language task, those stimuli would be left out during surgery. Intraoperatively, language stimuli were shown on an adjustable computer monitor that could be customized to each patient’s head position to increase the patient’s comfort as much as possible. Stimuli were displayed using Microsoft PowerPoint [33]. Every language trial was displayed for 4 s. For temporal coordination of the language stimuli presentation and DCS, a short acoustic tone was played at the beginning of each language trial to indicate to the surgeon to start the electrocortical stimulation. We used a negative-mapping technique with a limited and tailored craniotomy that exposed the tumor and a small margin of normal, peritumoral brain tissue. This approach allowed for stimulation mapping around the tumor margins but avoided extensive brain exposure only to force identifying all positive language sites. This technique was proven safe [34], minimized surgery time, and prevented complications of large skull flaps. During the language mapping with DCS, we used a rectangular biphasic pulse with a duration of 1.0 ms and a frequency of 50 Hz, bipolar electrodes for cortical stimulation and a monopolar electrode for subcortical stimulation, and currents varying between 1 and 6 mA. If a language disturbance referring to a single DCS locus was detected, the disturbance in association with the same stimulation locus had to be verified at least two more times. In turn, this stimulation site was defined as “language critical.” Possible occurring language problems or deficits according to DCS were classified and documented in all 24 patients. Additionally, language-positive and -negative points were recorded using the Brainlab navigation system [35].

## 3. Results

### 3.1. Evaluation Study

The mean performance duration for the language paradigm was 2.53 s in healthy subjects, with a standard deviation of 0.39 s. Overall, only 2 of 80 language stimuli led to processing times that exceeded 3 s (*M* = 3.17, *SE* = 1.22 and *M* = 3.31 s, *SE* = 1.1 s, respectively).

According to the fMRI experiment, we investigated which brain areas showed associated activation according to the language paradigm introduced in this study. The language paradigm led to predominantly left-hemispheric activations in the opercular, triangular, and orbital part of the inferior frontal gyrus, the putamen, the pallidum, the pre-and postcentral gyri, the hippocampus, and the inferior parietal gyrus (*T* = 7, *p* < 0.001; uncorrected). Moreover, we observed activation in the right hemisphere in the lingual gyrus, the angular gyrus, and the medial temporal gyrus. There was also bilateral enhanced activation in the fusiform gyrus, the occipital medial, inferior and superior gyri, the inferior temporal and superior parietal gyri, the thalamus, the supplementary motor area, and the insular cortex (*T* = 7, *p* < 0.001; uncorrected) (Figure 2, Table 2). Lowering the statistical threshold of *T* = 5 at the voxel level, which was still assumed to be a very conservative statistical threshold, there was also bilateral activation of the posterior superior and middle temporal gyri (*p* < 0.001; uncorrected).

### 3.2. Feasibility Study

Preoperatively, all patients showed average memory and visuo-constructive skills. For verbal working memory functions, the mean percentile ranks (*PR*s) for immediate memory access were *PR* = 22.2 (*SD* = 28.0), for delayed memory access *PR* = 14.2 (*SD* = 23.7), and for memory recognition *PR* = 29 (*SD* = 25.4). Visual memory recognition reached a *PR* = 35.2 (*SD* = 28.8) and visuo-constructive functions reached a *PR* = 31.5 (*SD* = 21.1). 

Intraoperatively, all patients (*N* = 24), even if they exhibited slight cognitive or language presurgical deficits, were able to perform the novel language paradigm after being intensively trained on the day prior to surgery. Language problems or deficits during language testing and the DCS could be observed in all 24 patients. Error characteristics were word-finding errors in 29% of patients, phonematic paraphasia in 11% of patients, semantic paraphasia in 5% of patients, morphological errors in 11% of patients, and syntactic errors in 13% of patients. A total of 11% of patients exhibited dysarthria, while also 16% of patients showed unspecific speech arrest (Table 1, Figure 3).

Postoperatively, due to general qualitative clinical observations, language skills were not affected by surgical intervention in 63% of patients, improved compared to presurgical language skills in 33% of patients, and declined in 4% of patients (*n* = 24) (Table 1). Thirteen patients did not differ significantly in their neuropsychological profile except for the semantic fluency task, which showed significantly lower percentile ranks after surgery with a high effect size (*p* = 0.015, *r* = 0.7). The effect sizes were medium for lexical fluency (*r* = 0.4) and weak for verbal working (number repetition forward (*r* = 0.2)), for number repetition backward (*r* = 0.2), for nonverbal working memory (*r* = −0.05), and for executive function tests (Trail-Making A: *r* = 0.01; Trail-Making B: *r* = 0.02) (Figure 4).

## 4. Discussion

The main aim of this study was to design a single novel intraoperative language paradigm that combined the advantages of the two most applied language tasks, which are picture naming and verb generation, to assess highly relevant language functions. Therefore, we first analyzed whether the language task led to valid activation in language associated areas around the sylvian fissure in a healthy subject sample in an fMRI experiment. The results showed language-paradigm-associated activation in a widespread network, including the inferior frontal and premotor areas, as well as the inferior parietal and posterior temporal areas in the left (and language dominant) hemisphere. Those areas are described as highly language critical in the current approaches of neural language processing [36,37,38]. Current models of language representation and processing suggest that successful natural language utterance is necessarily composed of semantic, lexical, and syntactic/grammatic information [39,40,41]. During object naming alone, mainly semantic and lexical properties of language are processed, while verb generation per se mainly requires lexical and grammatical information processing. Even if syntactic processing might be triggered through verb generation (e.g., because of the verb-associated valence structure and theta roles), it does not have to be executed. Combining both tasks demands the linguistic processes required by both tasks and requires the application of syntactic processes.

Besides the linguistic validity of the task, a single trial of the language task was supposed to be performed within 4 s to minimize the risk for intraoperative complications, such as seizures. The results of healthy subjects showed that the task could be performed in less than 3 s in all stimuli combinations, except for two trials, which took slightly more than 3 s to be conducted. This is in line with the standard safety protocols during DCS [24]. The performance speed of 3 s also allowed for some interindividual variance, and therefore, could also be applied in older patients, children, or even patients with persisting mild speech and language disturbances, such as decelerated speech or language.

Finally, the novel language task was also used during awake surgery in 24 patients with brain tumors in different language-critical locations in the frontal, parietal, and temporal lobes. All patients could perform the task without problems. Even if the language task performance was simple, all patients produced a variety of speech errors (Figure 3) under DCS, which could be characterized as either semantic, lexical, or syntactic/grammatic language processing errors, and therefore reflected language capacity with all its subfunctions. Of course, different errors occurred depending on the different tumor locations. Inferior parietal tumor patients produced errors that could be characterized as dysfunctional phonological retrieval. This means phonological information could not be assigned as semantic and possibly grammatical or syntactical [42]. Patients with inferior frontal or temporal tumors showed a higher variety and complexity of language errors. There are early reports of adapting intraoperative language testing to the tumor location. Duffau et al. argued for a specific task according to inferior parietal tumors (calculation tasks) and tumors in the middle temporal lobe (semantic and repetitions tasks) [43,44]. Bello et al. specifically applied counting and naming tasks for frontal tumors, as well as word and sentence comprehension tasks for temporal lesions [10,12]. Recent reviews report different language errors in association with different brain tumor locations [19,21]. Therefore, these authors suggest sets of different language tasks that are individually adapted and combined depending on the tumor location. However, this might be correlated with increased surgery times and affect patients’ attention and performance [22]. Therefore, de Witte and Marien postulate intraoperative language paradigms that provide maximum information gain in minimum exposure time to achieve successful valid intraoperative language monitoring [4]. Until now, there is no consensus or standardization with respect to this issue and intraoperative language monitoring per se, which is highlighted by Sefcikova and colleagues, who claim that the high interobserver, interinstitutional, and interspecialty variability of intraoperative language monitoring affects the validity, interpretation, and predictive power of intraoperative mapping [45].

According to postoperative clinical examinations, all patients with one exception showed stable or even improved linguistic performance after the surgery. Postoperatively, no patient was newly diagnosed as aphasic. One patient exhibited dysarthric symptoms (see Table 1). More sensitive neuropsychological testing revealed that patients did not significantly worsen in verbal and non-verbal working memory or executive functions. However, there was a significant decline in semantic verbal fluency. The lexical verbal fluency performance was not statistically significant between tests conducted before and after surgery; however, there was a trend of a slight decline in performance, as indicated by the medium effect size according to the statistical comparison. Looking closer at the data, this effect was mainly caused by two patients. Both had a low cognitive profile even before surgery, which might be predictive of postsurgical outcomes. One of these two patients also had a postsurgical hemorrhage. Verbal fluency tasks are in general less language-specific and reflect more general cognitive processes, such as neural inhibition and cognitive flexibility [46]. The outcome difference in the verbal fluency task could not be attributed to the surgical event alone because patients did not deteriorate in other cognitive domains (effect sizes according to statistical comparisons were small for these subtests). However, there might be an impact of tumor location and surgery accompanying events such as an edema. This might lead to subtle transient language difficulties because all tumors were in language-critical areas and had neuropsychological testing only a few days after surgery. Reports of pre- and postoperative cognitive and linguistic profiles after awake surgery are not coherent. Some authors report short- and long-term worsened linguistic but also cognitive skills [47,48,49]. However, there are also reports of only mild performance differences between pre- and postoperative linguistic and cognitive profiles regarding awake surgery [50,51]. Moreover, the results of a recent meta-analysis point out that functional testing and concordant DCS even minimizes the risk for long-term neurological and language deficits [1].

There are limitations to the data presented in this study. First, we could only report outcome neuropsychological data of a subsample (*n* = 13) of the whole patient group (*N* = 24). Even if this was done randomly because of limited data availability, we could not completely exclude a selection bias. Second, the patients’ preoperative cognitive profile was highly variable, which was reflected in high standard errors, and might therefore have influenced the intraoperative results and the postoperative outcome. Even still, the cognitive profile was sufficient in all patients, as shown by their ability to perform the intraoperative task. Third, the sample sizes of *N* = 24 for intraoperative feasibility of the language paradigm, as well as the sample size of *n* = 13 for the pre- and postoperative neuropsychological comparison, were rather small. Larger sample sizes would be necessary to replicate and confirm the results presented in this study. Fourth, a direct comparison between the novel language task applied in this study and other language task setups (e.g., picture naming or verb generation) or a combination of tasks is missing in terms of surgery time, the incidence of intraoperative seizures, and the extent of resection. Therefore, we can only assume that the application of the novel language tasks shortened surgery duration and improved patients’ intraoperative attention. A comparison with other tasks and paradigms should be the topic of further research. Fifth, even if the application of the novel language task can safely monitor most of the highly relevant language functions by using one single task, it is nevertheless not possible to assess the entire language system with a single task. The paradigm might be insufficient for processes such as semantic categorization or more complex syntactic operations. Sixth, we applied the novel language paradigm during pre-surgical functional imaging in healthy subjects and not in patients. Therefore, we can make no statements regarding how brain activation associated with the novel language paradigm would appear in the patient groups. Moreover, this would demand a group stratification due to tumor location; however, our sample size was insufficient for this. Furthermore, we could not assess the validity of the novel language task according to the comparison of presurgical fMRI and intraoperative language testing during DCS. However, these issues were not in the focus of this study and exceeded its scope. As far the paradigm was only applied to native German speakers, we do not know whether the translation in other languages may influence performance speed because of different language-specific syntactical and morphological requirements. Moreover, it is unclear whether the novel language paradigm is appropriate in bilingual subjects. Taken together, this study does not claim to establish a new gold standard for intraoperative language mapping. Instead, we took a closer look at the feasibility and validity of our novel task.

## 5. Conclusions

The novel intraoperative language paradigm introduced in this study proved to be feasible and safe during intraoperative language mapping. Consisting of only a single language task, it depicted highly relevant language functions, allowing for more efficient intraoperative testing than object naming or verb generation alone, and thereby may improve patients’ intraoperative testing compliance. This is especially important for even slightly cognitively and linguistically impaired patients or pediatric patients. For both groups, one might consider awake surgery very carefully since these patients could be overburdened regarding the surgery duration and the cognitive demands of a battery of intraoperative language tasks. Additionally, the application of this test may improve postoperative linguistic and cognitive outcomes.

## Figures and Tables

**Figure 1 jcm-10-00655-f001:**
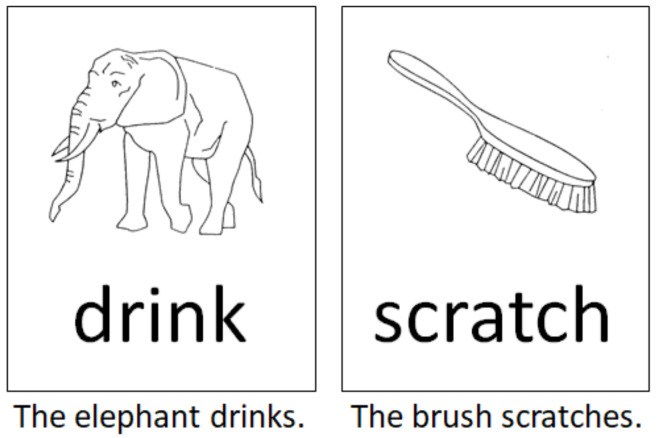
Two examples according to the intraoperative language task used in this study. Subjects and patients were asked to generate grammatically and semantically correct sentences according to the language stimulus as depicted.

**Figure 2 jcm-10-00655-f002:**
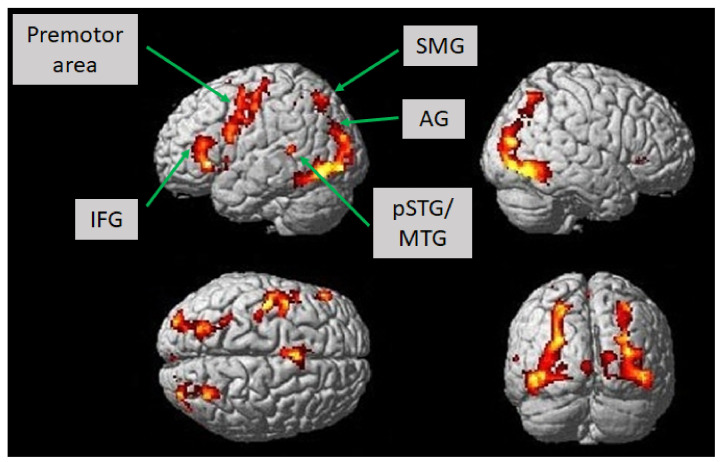
Brain activation in healthy subjects (*n* = 12) while constructing grammatically and semantically correct sentences in response to the novel language task, which comprised a combination of picture naming and simple verb generation (*T* = 7, *p* < 0.001; uncorrected). Activation is depicted on a single-subject-normalized brain surface implemented in SPM8. Abbreviations: IFG—inferior frontal gyrus, SMG—supramarginal gyrus, AG—angular gyrus, pSTG—posterior superior temporal gyrus, MTG—middle temporal gyrus.

**Figure 3 jcm-10-00655-f003:**
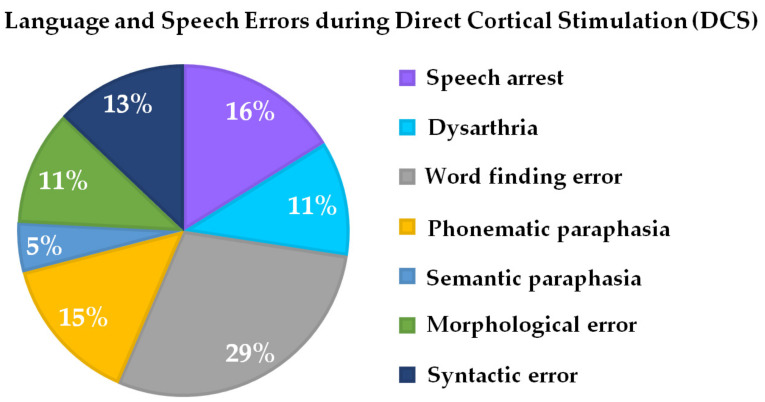
Percentage of intraoperative language and speech errors during DCS associated with the novel language paradigm (see Figure 1).

**Figure 4 jcm-10-00655-f004:**
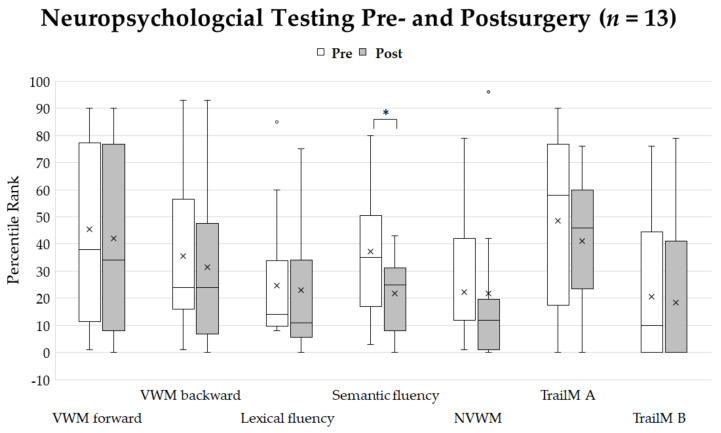
Percentile ranks of neuropsychological testing before and after surgery. Crosses show the mean values. Asterisk shows significant differences of percentile ranks (*p* < 0.05). The only significant difference was seen for semantic verbal fluency. Abbreviations: VWM-visual working memory, NVWM-non-visual working memory, TrailM A-Trail-Making Test A, TrailM B-Trail-Making Test B.

**Table 1 jcm-10-00655-t001:** Patients characteristics, including pre-, intra-, and postsurgical language deficits.

Patient Number	Age	Handedness	Tumor Location	Diagnosis	Presurgical Karnofsky	Presurgical Seizures	Presurgical Deficit	Postsurgical Deficit	Intraoperative Deficit
1	46	Right	IF	OD II, recurrent	100	Yes	None	None	WF, SA, PP, SP, SE, ME
2	21	Right	SG, AG	AA III	90	Yes	Mild AP	Mild AP	WF, SA
3	61	Right	SG, AG	GBM	100	Yes	None	None	WF, SA
4	28	Right	IF	A II	100	Yes	None	None	WF, SA
5	47	Right	IF	AOD III	100	Yes	None	Mild DA	SA, DA
6	26	Right	STG	AA III	100	Yes	None	None	WF, SE, ME
7	36	Right	STG, MTG	AA III	90	No	WF	None	DA, PP
8	38	Right	Premotor area	AA III	100	Yes	None	None	WF, SA, DA, ME
9	32	Right	STG, MTG	AA III, recurrent	90	Yes	Mild DA	None	WF, PP, SE, ME
10	27	Right	IF	AA III, recurrent	100	No	None	None	WF, PP, SE
11	27	Left	IF	AA III	100	Yes	None	None	WF
12	37	Right	Premotor area	AA III	100	No	None	None	SA
13	52	Right	IF	AA III	100	No	None	None	WF, DA, PP, SE, ME
14	55	Right	SG	AA III	90	Yes	Mild DA	Mild DA	DA
15	68	Right	IC, premotor area	AA III	90	Yes	Mild AP	None	WF, SA
16	76	Right	IF	AOD III	100	No	None	None	WF, PP
17	12	Right	MTG	GG I	90	Yes	WF	None	WF, SA, PP, SP
18	20	Right	STG	PA	90	Yes	WF	None	WF
19	53	Right	Premotor area	GBM	90	Yes	Mild DA	Mild DA	WF, ME, SE
20	59	Right	SG, postG	GBM	90	Yes	WF	Mild DA	DA
21	41	Right	IF	AA III	100	Yes	None	None	WF, SA
22	61	Right	IF	AA III	90	Yes	Mild DA	None	WF, DA, PP, ME, SE
23	27	Right	IF	AA III	100	Yes	None	None	WF
24	42	Right	MTG	OD II	100	Yes	None	None	PP, SP, SE

Abbreviations: WFword finding error, AR—speech arrest, DA—dysarthria, PP—phonematic paraphasia; SP—semantic paraphasia, ME—morphological error, SE—syntactic error, AP—aphasia, OD—oligodendroglioma, AOD—anaplastic oligodendroglioma, GBM—glioblastoma, A—astrocytoma, AA—anaplastic astrocytoma, GG—ganglioglioma, PA—pilocytic astrocytoma, IF—inferior frontal, SG—supramarginal gyrus, AG—angular gyrus, STG—superior temporal gyrus, MTG—middle temporal gyrus, PostG—postcentral gyrus, IC—insular cortex.

**Table 2 jcm-10-00655-t002:** Brain activation in healthy subjects (*n* = 12) while constructing grammatically and semantically correct sentences according to the language task, which comprised a combination of picture naming and simple verb generation (*T* = 7, *p* < 0.001; uncorrected).

MNI Coordinates			Cluster Size	Region	Brodmann Area	Hemisphere	*T*-Value	*p* _FWE_
*x*	*y*	*z*						
34	−86	18	2598	Fusiform gyrus, MOG, IOG, SOG, lingual gyrus, SPG, MTG	7, 18, 19, 37	R	21.85	0.000
−18	2	8	342	Putamen, pallidum		L	17.87	0.000
−38	2	34	835	Precentral gyrus, postcentral gyrus, IFG	4, 6	L	17.46	0.000
−8	−24	−10	881	Thalamus, hippocampus		L	16.31	0.000
−40	−84	−4	2497	MOG, fusiform gyrus, IOG, SPG, ITG, IPG, SOG	7, 19, 37	L	16.16	0.000
24	−30	−2	268	Thalamus		R	13.35	0.000
−6	14	44	424	SMA	6	L and R	13.19	0.000
−24	26	12	541	IFG (PT), IFG (PO), Insula		L	10.80	0.000
32	28	−2	96	Insula		R	9.59	0.000
−2	−94	2	50	Fissura calcarina		L	8.97	0.000

Abbreviations: MOG—middle occipital gyrus, SOG—superior occipital gyrus, SPG—superior parietal gyrus, MTG—middle temporal gyrus, ITG—inferior temporal gyrus, IFG—inferior frontal gyrus, PT—pars triangularis, PO—pars opercularis, IFG—inferior parietal gyrus, IOG—inferior occipital gyrus, SMA—supplemantary motor area, L—left, R–right, MNI—Montreal Neurological Institute, FWE—family-wise error. Anatomical labeling was done using the Automated Anatomical Labeling (AAL) atlas implemented in SPM8.

## Data Availability

The data presented in this study are available on request from the corresponding author. The data are not publicly available due to patients’ privacy.

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
