# Peer review of "A Novel Language Paradigm for Intraoperative Language Mapping: Feasibility and Evaluation"

_jcm, 2021, doi:10.3390/jcm10040655_

Round 1

Reviewer 1 Report

The manuscript entitled “A novel language paradigm for intraoperative language mapping: feasibility and evaluation” presents a novel language task for intraoperative language mapping. The authors make a valid point that using a single language task intraoperatively may not be sufficient to map language functions adequately. It is also true that awake surgeries have their time limitations, both in terms of overall language mapping duration, as well as single stimulus presentation. Therefore, new language paradigms that can address these challenges are highly important to neurosurgery and clinical neuropsychology. The novel language task introduced in the manuscript can be a useful addition to intraoperative language mapping in the future. However, the are several major and minor issues that limit the manuscript:

MAJOR ISSUES

  1. Abstract: The authors state that the new language task engages the entire language system. This is an overstatement that appears throughout the paper. As detailed below, using fMRI in healthy individuals, the task produced scarce activations in the posterior superior and middle temporal gyri. These regions constitute critical language areas that need to be preserved post-surgically. For example, it is known from the literature (e.g., Dronkers et al. 2004) that the posterior middle temporal gyrus is an essential region for grammar processing.
  2. Page 2, line 48. Grammar, syntax, and morphology are incorrectly listed as different language aspects. Grammar is a collective term that encompasses syntax and morphology. Thus, one could say “grammar (syntax and morphology)”. This mislabeling occurs multiple times throughout the manuscript.
  3. Page 2, lines 61-61. The authors suggest that “naming tasks alone should be abandoned especially in patients with frontal lesions". It is a risky statement to make in neurosurgery, and the current paper (as presented) does not offer a convincing alternative to the naming tasks. The object naming task is considered the gold standard of intraoperative language mapping. Many neurosurgeons claim that they successfully use these tasks in all their patients, which is backed by the literature. What the authors may want to say instead is that it may be helpful to increase the sensitivity of the object naming task by adding a grammatical component to it. Such addition may increase the likelihood of detecting eloquent language regions that may currently be overlooked using lexico-semantic tasks.
  4. A separate Subjects section should be added to the Methods with the three groups of participants.
  5. Methods, p. 3: It is not clear whether, in the second group of 12 healthy participants, there were any individuals who took part in the first study (30 subjects). If yes, how much time passed between the first and the second study? How did the authors account for a potential familiarity effect of the presented stimuli?
  6. Page 3: The sample size of the healthy participants in the fMRI experiment is small (n=12).
  7. Page 3, lines 121-122:The frontal operculum and the inferior frontal gyrus were listed among the language sites that the new language task could activate. The frontal operculum is part of the inferior frontal gyrus, not a separate region.
  8. Page 4, line 158: Pre-surgical fMRI in patients should be introduced earlier as it comes as a surprise at this point. I suggest reorganizing the Methods section into the more standard components, including the previously mentioned Subjects section, but also a Procedures section in which all the experiments are described.
  9. Methods: one of my main problems with this study is that the fMRI results for the novel language task are provided only for 12 healthy participants, whereas intraoperative mapping results for the task come from 24 patients. While the fMRI results from the healthy controls are essential to understand which brain sites the new tasks can activate, we do not know how similar or different these activations are in the 24 patients with brain tumors residing in the language network. Moreover, we also do not know how the fMRI results in patients (either from the new task or standard language tasks) correspond to the findings from the awake language mapping.
  10. Methods: It would be informative to see a map of all the eloquent language sites identified using the novel task during intraoperative language mapping. Such a map would show the extent of language sites detected during the task. The authors provide no specifics as to which language sites were detected intraoperatively with the task, which is another major limitation of this work.
  11. Page 6, Results. Figure 2: Based on the fMRI results in the healthy subjects, it appears that the new language task generated few activations in the left posterior middle and superior temporal gyri, which is concerning for patients with lesions within these areas. Activations in these two regions were not listed in Table 1, meaning that they were likely insignificant. Considering the lack of activation in these regions, the author’s statement that their task engages the entire language system does not seem to be correct.
  12. Page 6, lines 243-245: Linking back to comment no. 2 above, it is not clear which language errors the authors included under “morphological”, “syntactic”, and “grammar” errors. The authors provided examples for grammar errors (“verb flection errors”) but these are morphological errors.
  13. Page 7, Table 2: There was one pediatric patient (age 12) in the sample. I would suggest excluding this patient from the sample. This study aims to examine the ability of a new language task to identify eloquent language sites in the brain that isfully developed by the time of admittance of patients to the clinic, not when brain plasticity is significantly more robust (early teenage years).
  14. Page 7, Table 2: The authors used the term "word-finding disorder" in the context of intraoperative language mapping. Since language disturbances induced by stimulation mapping are transient, the term "word-finding error” would be more appropriate.

MINOR ISSUES

  1. Page 2, line 48: In the sentence “(…) organized as a complex network of brain areas and fiber bundles (…)”, the authors probably meant to say “cortical areas”.
  2. There are occasional language errors in the manuscript that need editing, e.g., “increasing surgery times” (it should say “increasing surgery time”), “at the begin of each language trial” (it should say “at the beginning of each language trial”).
  3. Page 3, line 157: instead of “wake-awake-awake”, it should say “sleep-awake-sleep”.

Author Response

Response to Reviewer 1 Comments

  1. Abstract: The authors state that the new language task engages the entire language system. This is an overstatement that appears throughout the paper. As detailed below, using fMRI in healthy individuals, the task produced scarce activations in the posterior superior and middle temporal gyri. These regions constitute critical language areas that need to be preserved post-surgically. For example, it is known from the literature (e.g., Dronkers et al. 2004) that the posterior middle temporal gyrus is an essential region for grammar processing.

Response:

Thank you for this valuable comment. We weakened the statement “entire” with “highly relevant language functions”.

The reviewer is right, that the task shows less activation in posterior temporal areas than for example in frontal or occipital areas. This is due to statistical and paradigm associated issues.

The language task introduced in this manuscript is a very complex fMRI task resulting in the activation of language critical areas but also non-essential regions that support the language network without being language-critical. These language-supporting areas reflect functions and cognitive processes like visual-perception, working memory, memory, attention, object recognition etc. Because of the emphasis on general activation patterns of the new task an elimination of these processes by using (linguistic) control conditions was not appropriate in this study.

However, this approach requires the use of very conservative p-values. As stated in the text, lowering the statistical threshold (line 243) (on still very reasonably T-values) results in activation in posterior and superior temporal areas.

Another issue relates to interindividual spatial variance of these posterior brain regions. This leads to further problems regarding the detection of activated clusters within the 2-level analysis. Particularly, as the individual analysis (1-level) revealed reliably activation in posterior and superior temporal areas.

  1. Page 2, line 48. Grammar, syntax, and morphology are incorrectly listed as different language aspects. Grammar is a collective term that encompasses syntax and morphology. Thus, one could say “grammar (syntax and morphology)”. This mislabeling occurs multiple times throughout the manuscript.

Response:

We thank the reviewer for this clarification. It is definitely correct that grammar encompasses combination rules for syntax and morphology but according to official definitions even to phonology and semantics. We tried to address this issue by skipping the misleading term “grammar” and add the according deficits to the categories of syntax and morphology which is more or less appropriate.

  1. Page 2, lines 61-61. The authors suggest that “naming tasks alone should be abandoned especially in patients with frontal lesions". It is a risky statement to make in neurosurgery, and the current paper (as presented) does not offer a convincing alternative to the naming tasks. The object naming task is considered the gold standard of intraoperative language mapping. Many neurosurgeons claim that they successfully use these tasks in all their patients, which is backed by the literature. What the authors may want to say instead is that it may be helpful to increase the sensitivity of the object naming task by adding a grammatical component to it. Such addition may increase the likelihood of detecting eloquent language regions that may currently be overlooked using lexico-semantic tasks.

Response:

The reviewer raises an important point, and we agree that our suggestion to abandon naming tasks goes too far. In particular, our study does not claim to establish a new gold standard. Instead, we took a closer look on feasibility and validity of our novel task. A comparison to other tasks and paradigms should be the topic of further research. Therefore, we adjusted the text according to the reviewer’s advice and marked the appropriate text (line 60).    

  1. A separate Subjects section should be added to the Methods with the three groups of participants.

Response:

We thank the reviewer for this helpful suggestion. Therefore, we changed the structure of the methods section and included a “subject section” to increase the manuscript’s readability and comprehensibility.

  1. Methods, p. 3: It is not clear whether, in the second group of 12 healthy participants, there were any individuals who took part in the first study (30 subjects). If yes, how much time passed between the first and the second study? How did the authors account for a potential familiarity effect of the presented stimuli?

Response:

The reviewer is right. Unfortunately, this point was not described in detail.

The 12 subjects of the fMRI experiment were different from those 30 subjects of the behavioral task. This is now clarified and marked in the text (line 123).

  1. Page 3: The sample size of the healthy participants in the fMRI experiment is small (n=12).

Response:

We thank the reviewer for this consideration which needs to be discussed.

In our opinion, 12 healthy participants for a fMRI-experiment on the basis of our methodological conditions are acceptable. Apart from the fact that an ultimate determination of a sample size reflecting predefined statistical power requires a sample size calculation. Furthermore, the fMRI's inherent repeated-measurement design and our analysis considering the hierarchical structure of the data increase statistical power enormously.

  1. Page 3, lines 121-122: The frontal operculum and the inferior frontal gyrus were listed among the language sites that the new language task could activate. The frontal operculum is part of the inferior frontal gyrus, not a separate region.

Response:

We thank the reviewer for the correction, and we are sorry for possible confusing nomenclature.

Therefore, we replaced “frontal operculum” with “ventral premotor area” in the text.

  1. Page 4, line 158: Pre-surgical fMRI in patients should be introduced earlier as it comes as a surprise at this point. I suggest reorganizing the Methods section into the more standard components, including the previously mentioned Subjects section, but also a Procedures section in which all the experiments are described.

Response:

We thank the reviewer for this suggestion and restructured the methods section to improve the readability of the manuscript by organizing the section in a subject and a procedures section.

Therefore, we switched Table 1 and 2. Table 1 now shows the subjects’ data and Table 2 the fMRI results in healthy subjects. The Pre-surgical fMRI is now early mentioned in the subjects section.

  1. Methods: one of my main problems with this study is that the fMRI results for the novel language task are provided only for 12 healthy participants, whereas intraoperative mapping results for the task come from 24 patients. While the fMRI results from the healthy controls are essential to understand which brain sites the new tasks can activate, we do not know how similar or different these activations are in the 24 patients with brain tumors residing in the language network. Moreover, we also do not know how the fMRI results in patients (either from the new task or standard language tasks) correspond to the findings from the awake language mapping.

Response:

The reviewer raises an important point. However, the main aim of the study was to introduce a suitable language paradigm in the setting of awake surgery. It is important to consider the specific requirements of tasks for intraoperative testing in contrast to fMRI-examination. In our opinion, a good intraoperative language task should be rather linguistic complex so that it is predictive for a wide range of potential post-operative language deficits. The utmost goal of the fMRI-procedure is to reveal the localization of language areas by reliable activation of language critical brain areas and to fade out non-language critical processes. Therefore, it is not necessary that fMRI tasks must be complex. Instead, there is a need for the use of different language task and appropriate statistical analysis of the data. This in mind, our intention was not to evaluate and validate the new language task for the use in pre-surgical functional imaging, particularly as the patient group did not perform the new language task within the fMRI-examination. Of course, we applied a valid battery of fMRI-tasks including word and sentence generation tasks for the determination of language lateralization as well as localization of anterior and posterior language-critical areas. Therefore, the presentation of these results would only enrich the sample description. In addition, due to different tumor locations it is not reasonable to perform group analyses.

To sum up, the aim of the study was not to validate any fMRI data by the criterion to intra-operative electro-cortical stimulation. Furthermore, this would not be possible because of different language paradigms during fMRI and surgery. We also added a considerate comment on this in the limitation section (line 397).

  1. Methods: It would be informative to see a map of all the eloquent language sites identified using the novel task during intraoperative language mapping. Such a map would show the extent of language sites detected during the task. The authors provide no specifics as to which language sites were detected intraoperatively with the task, which is another major limitation of this work.

Response:

We thank the reviewer for this important consideration. We wanted to investigate if the novel language task can be performed during awake surgery in terms of feasibility. In addition, the benefit was determined by comparing neuropsychological performance before and after surgery. Furthermore, it would be definitely highly informative to see the intraoperative language sites.

We used a negative-mapping technique with a limited and tailored craniotomy exposing the tumor and a small margin of normal, peritumoral brain tissue. This approach allows stimulation mapping around the tumor margins but avoids extensive brain exposure only to force identifying all positive language sites. This technique was proven safe, minimizes surgery time and prevents complications of large skull flaps*. Against this backdrop, it is not possible to show the complete extent of language sites detected during the task at least on a single subject level.

We added this information in the methods section.

* Sanai N, Mirzadeh Z, Berger MS. Functional outcome after language mapping for glioma resection. N Engl J Med. 2008;358:18-27.

  1. Page 6, Results. Figure 2: Based on the fMRI results in the healthy subjects, it appears that the new language task generated few activations in the left posterior middle and superior temporal gyri, which is concerning for patients with lesions within these areas. Activations in these two regions were not listed in Table 1, meaning that they were likely insignificant. Considering the lack of activation in these regions, the author’s statement that their task engages the entire language system does not seem to be correct.

Response:

We commented on this issue already earlier in comment 1.. Additionally, we weakened the statement “entire” with the more appropriate term “highly relevant language functions”. We also added a short paragraph according to that issue in the limitation section (line 394).

  1. Page 6, lines 243-245: Linking back to comment no. 2 above, it is not clear which language errors the authors included under “morphological”, “syntactic”, and “grammar” errors. The authors provided examples for grammar errors (“verb flection errors”) but these are morphological errors.

Response:

We thank to reviewer for this worthful advice. We also commented on this in 2. Please see above.  We skipped the category of grammar errors and assigned those errors to either the syntactic category (mainly representing grammatical gender or declination errors) or the morphological error category (which mainly comprise verb flection errors). We exchanged and marked this in the text of the result section, in Table 1, and the Figure 3.

  1. Page 7, Table 2: There was one pediatric patient (age 12) in the sample. I would suggest excluding this patient from the sample. This study aims to examine the ability of a new language task to identify eloquent language sites in the brain that is fully developed by the time of admittance of patients to the clinic, not when brain plasticity is significantly more robust (early teenage years).

Response:

We agree with the reviewer that language is a function which exhibits huge plasticity until late teenage years.

In this study only intraoperative language performance data of the pediatric patient was included where we do not expect performance differences of a young teenager and adults according to the task. The task which requires only low cognitive skills next to language is trained intensively before surgery as stated in the text.

Of course, it would not be appropriate to compare e.g. neuropsychological data or fMRI data which we both did not do. Patient’s fMRI data were not included in the study as well as the pediatric patient’s neuropsychological data (here even the tests where different form the adult ones).

  1. Page 7, Table 2: The authors used the term "word-finding disorder" in the context of intraoperative language mapping. Since language disturbances induced by stimulation mapping are transient, the term "word-finding error” would be more appropriate.

Response:

We thank the reviewer for this comment and will exchange “disorder” with “error” in all places.

MINOR ISSUES

  1. Page 2, line 48: In the sentence “(…) organized as a complex network of brain areas and fiber bundles (…)”, the authors probably meant to say “cortical areas”.

Response:

We thank the reviewer for this information and changed the wording in the text.

  1. There are occasional language errors in the manuscript that need editing, e.g., “increasing surgery times” (it should say “increasing surgery time”), “at the begin of each language trial” (it should say “at the beginning of each language trial”).

Response:

We thank the reviewer for these comments and changed the wording in the text.

  1. Page 3, line 157: instead of “wake-awake-awake”, it should say “sleep-awake-sleep”.

Response:

We thank the reviewer to address this issue. Our institution exclusively uses the “wake-awake-awake” anesthesia protocol, which was published by Hansen et al.*. This means patients are awake all throughout the surgery (also e.g. during local anesthesia or craniotomy) without any sedation in order to affect cognitive capacity as less as possible. We are aware that this management is rather rare.

*E. Hansen, M. Seemann, N. Zech, C. Doenitz, R. Luerding, and A. Brawanski, “Awake craniotomies without any sedation: the awake-awake-awake technique,” Acta neurochirurgica, vol. 155, no. 8, pp. 1417–1424, 2013, doi: 10.1007/s00701-013-1801-2.

Reviewer 2 Report

This is an interesting manuscript that proposes to investigate a novel language assessment technique for intraoperative tumor resection of the dominant hemisphere.  The authors note that various language assessment techniques exist for intraoperative tumor surgery, yet these all have "blind spots" that may not adequately assess certain components of the visual/speech pathways, and thus may increase the potential of postoperative deficits.  

The authors describe a novel technique that involves a picture and a verb, and the patient is required to identify the picture and then include it in a sentence with correct syntax and verb generation.  The preliminary studies in controls with fMRI showed that language paradigm associated activation in a widespread network including inferior frontal an premotor areas, as well as inferior parietal and posterior temporal language critical areas in the dominant hemisphere.

The authors' technique of DCS with the flash cards was able to be performed in less than 3 seconds, during which time language arrest could be tested with DCS. 

Utilization of the authors' technique led to almost all patients having improved linguistic performance after surgery. 

Overall, this study shows an interesting novel technique of language assessment that was evaluated with fMRI and introperative studies and found to be a good technique for testing multiple components of language functioning at once, to provide more accurate and safe tumor resection in tumors in the dominant hemisphere.

Author Response

Response to Reviewer 2 Comments

We thank the reviewer for the encouraging and supporting words regarding our study.

Reviewer 3 Report

This is a paper based on a very interesting idea of optimizing the evaluation of language mapping during awake craniotomy. However the usefulness and impact on surgical results are vague and yet to be assessed. 

The authors might consider rewriting the abstract to clarify the aims of the study. I also recommend the authors to clarify the term "linguistic subprocesses" (line 64).

The authors also describe that the patients were intensively trained in advance to surgery. Doesn't this cause a bias and make the results difficult to interpret? Do the authors suggest that similar training would be used in a clinical setting? 

What limitations does the new paradigm have in regards to patients with impaired functions, bi-lingual patients or individuals who are not native speakers of the language?

The introduction to the Discussion is very good an clarifies a lot and the authors make an honest account of the limitations of the study. However, the reasoning behind the authors assuming that the test improves surgical setting is somewhat unclear. It would also be interesting if the authors commented on whether they think this paradigm might change surgical decision making.

Author Response

Response to Reviewer 3 Comments

  1. This is a paper based on a very interesting idea of optimizing the evaluation of language mapping during awake craniotomy. However, the usefulness and impact on surgical results are vague and yet to be assessed.

Response:

We thank the reviewer for this consideration and agree completely.

This study does not claim to establish a new gold standard for intraoperative language mapping. Instead, we took a closer look on feasibility and validity of our novel task. A comparison to other tasks and paradigms should be the topic of further research. We also mentioned that at the end of the limitation section (line 409).

  1. The authors might consider rewriting the abstract to clarify the aims of the study. I also recommend the authors to clarify the term "linguistic subprocesses" (line 64).

Response:

We agree with the reviewer, that the abstract needs clarification. We tried to reformulate the abstract. We thank the reviewer for the worthful suggestion and tried to specify “linguistic subprocesses” this in the text (line 67).

  1. The authors also describe that the patients were intensively trained in advance to surgery. Doesn't this cause a bias and make the results difficult to interpret? Do the authors suggest that similar training would be used in a clinical setting?

Response:

We thank the reviewer for this justified consideration. Of course, this is a bias but there are some reasons for it. First, we need to know if patients are able to conduct the task intraoperatively. For example, if patients are unfamiliar with single drawings we don´t use them during surgery. Or, it is also an important information if patients use dialect wordings or even grammar to judge language output intraoperatively. Second, for patients comfort it is very important that patients are familiar with the task in detail (as all other aspects of the intraoperative setting) because otherwise they might be overburdened. At least this intensive training next to other intensive presurgical preparation is done in every patient who is supposed to have awake craniotomy in our institution.

  1. What limitations does the new paradigm have in regards to patients with impaired functions, bi-lingual patients or individuals who are not native speakers of the language?

Response:

We thank the reviewer for this interesting issue. Some patients already had slight cognitive or even language deficits and were still able to perform the task (see Table 1, former Table 2). If patients are very strong impaired in their cognitive or language skills we would not suggest an awake craniotomy because intraoperative DCS-induced language deficits won’t be reliably interpreted.

We stated this now also in the text (line 272). We never applied the paradigm in bilinguals or not native speakers yet as we did neither with other language paradigms. Of course, one could try to adjust the paradigm in the relevant language, but this yet must be evaluated. Because, other languages might have different syntactical and morphological requirements, where we do not know if those can be managed in less than 4 seconds. Therefore, we added a short paragraph in the limitation section (line 404).

  1. The introduction to the Discussion is very good an clarifies a lot and the authors make an honest account of the limitations of the study. However, the reasoning behind the authors assuming that the test improves surgical setting is somewhat unclear. It would also be interesting if the authors commented on whether they think this paradigm might change surgical decision making.

Response:

The main advantage the novel language paradigm is shortening of the surgery time. The paradigm allows the testing of several highly relevant language functions at the same time. So, there is no need of an extensive test-battery of intraoperative tests. Especially, it seems to be appropriate to test several language functions in even slightly cognitively and linguistically impaired patients or pediatric patients (since one patient was 12 years old) reliably. For both groups and especially for younger patients one might not suggest awake surgery since these patients could be overburdened in means of surgery duration and cognitive demands by e.g. using more than one language task. We stated this also in the conclusion section (line 407).

Round 2

Reviewer 1 Report

The authors have significantly improved their manuscript. They have adequately addressed all of the Reviewers' comments.